# Does the Wolf (*Canis lupus*) Exhibit Human Habituation Behaviours after Rehabilitation and Release into the Wild? A Case Report from Central Italy

**DOI:** 10.3390/ani12243495

**Published:** 2022-12-11

**Authors:** Paolo Viola, Pedro Girotti, Settimio Adriani, Bruno Ronchi, Marco Zaccaroni, Riccardo Primi

**Affiliations:** 1Department of Agricultural and Forest Sciences, University of Tuscia, 01100 Viterbo, Italy; 2Department of Biology, University of Florence, 50019 Florence, Italy

**Keywords:** habitat selection, home range, human habituation, movement pattern, rescued wolf

## Abstract

**Simple Summary:**

The rehabilitation of injured or sick wolves is a practice that is undertaken until it is clinically possible to release the animals back into the wild. However, the knowledge of how movement patterns and habitat selection are eventually affected by habituation to persons after a period of veterinary isolation, treatment and non-agonistic experience with humans is scarce. We describe the behaviour of a radio-collared female wolf released into the wild after a rehabilitation period of 11 days. The wolf travelled about the same distances as wild conspecifics, showing movement patterns and circadian rhythms complementary to those adopted by humans. No signal of behavioural distortions due to human habituation were recorded. This case study aims to stimulate further research and a call for widespread data sharing at national and international scales.

**Abstract:**

The knowledge of how wolves’ movement patterns and habitat selection are affected by habituation to persons after a period of veterinary isolation, treatment and non-agonistic experience with humans is scarce. Unnatural behaviours could be transferred by imitation to members of the pack and to subsequent generations, increasing direct interaction risks. We used GPS data from a rescued radio-collared female wolf after an 11-day rehabilitation to estimate home range, movement patterns, circadian rhythms, and habitat selection, searching for signals of eventual behavioural distortions. In the period 1 August–26 November 2013, 870 valid locations were acquired. The wolf moved within a minimum convex polygon (95%) of 6541.1 ha (79% wooded), avoiding anthropized areas. Nocturnal and diurnal displacements were significantly different (*p* < 0.01). Nocturnal displacements were 4409.4 ± 617.5 m during summer and 3684.8 ± 468.1 m during autumn, without differences between seasons. Diurnal movements were significantly higher (*p* < 0.01) in the summer (2239.0 ± 329.0 m) than in the autumn (595.9 ± 110.3 m), when the hunting season was running. As for a wild wolf, clear complementarity concerning human activities was recorded and no habituation signals were detected, but this is only a first case study that aims to be a stimulus for further research and a call for widespread data sharing.

## 1. Introduction

The wolf (*Canis lupus* Linnaeus, 1758) is one of the most studied carnivores [1] because of its conservation value; its umbrella ecological functions [2]; and, recently, its negative interactions with human activities. Although predators are decreasing worldwide [3], the wolf in Europe recolonized many parts of their former range, leading to increased conflicts with anthropic interests [4,5,6], particularly with livestock farmers [7,8]. The wolf demographic and distributive expansion increases, as well as the opportunities of contact with free ranging dogs, imposing attention to the hybridization risk [9].

Currently, the Italian wolf population [10], which is genetically distinct from other European wolves, has increased from ca. 1500 individuals estimated in 2015 [4,11] to ca. 3307 (2945–3608) individuals estimated in 2021 [12], but despite this, the wolf is threatened, principally due to illegal kills [13,14,15]. Collisions with vehicles also represent one of the leading causes of wolf mortality [4,13,16] and can reduce population abundance by over 20% annually [11]. Injured wolves are rehabilitated and released into the wild when clinically possible [11]. However, very little attention has been given to the behaviour of these wolves when released into the wild after a period of veterinary isolation, treatment and non-agonistic experience with humans [11,17,18]. The topic deserves attention, since unnatural behaviours could increase the risk of post-release mortality, as well as be transferred by imitation both to the members of the pack and to subsequent generations, increasing the frequency of direct interactions and determining a further exacerbation of conflicts.

Aiming to start the advancement in knowledge on this specific topic, we used Global Positioning System (GPS) data from a female wolf, hereafter named Carlina, rescued at the east side of the Terminillo Mountain (province of Rieti, Italy) to estimate seasonal home range, movement patterns (distance and speed), circadian rhythms and habitat selection, searching for signals of eventual behavioural distortions, suggesting the occurrence of human habituation.

## 2. Materials and Methods

### 2.1. Study Area

The study area was located in the wide and mainly mountainous area of Central Apennine (Italy, Lazio region: Lat 42.4780 Long 12.93073) (Figure 1). In particular, the vast area overlaps a part of the Reatini Mountains group and the respective foothills flat area (Rieti plain), for a total surface of about 206 km^2^. The involved municipalities were Morro Reatino, Rivodutri, Posta, Cittaducale, Castel Sant’Angelo, Borgo Velino, Cantalice, Poggio Bustone, Rieti, Micigliano and Leonessa (Figure 1). It was partially interested by the SPA “Monti Reatini” (IT6020005), involving the protection area “Terminillo Oasis” and the regional natural reserve “Laghi Lungo e Ripasottile”, which also includes the homonymous SPA–SAC (IT6020011). Between the two protected areas, a hunting dog training zone of 197 ha was located. The study area has a minimum altitude of 380 m a.s.l. of the flat portion, and a maximum altitude of 2217 m a.s.l. of Terminillo mountain.

The climate is typical of the band temperate region. The central Apennines are characterized by the inferior subalpine thermotype with beech woods and shrubs, a *Juniperus alpina*, *Vaccinium myrtillus* and *Arctostaphylos uva-ursi*; by the lower mountain thermotype with beech woods and woods dominated by *Ostrya carpinifolia*; and by upper hill thermotype with hop hornbeam (*Ostrya carpinifolia*), mixed woods and oaks to *Quercus* spp. [19]. Forests cover over half of the territory, where deciduous and conifer woods represent 55.8% and 3.3%, respectively, while 13.4% is represented by natural grassland and pastures. Roads and human settlements cover 10% of the total study area (Table 1).

In the area, the wild boar was widely present and abundant, as well as the roe deer [20], and about 3116 ha of wild boar drive hunting zones were intersected. In these zones, drive hunting with dogs was performed in the period October–January, in accordance with national and regional hunting rules.

### 2.2. Application and Setting of the GPS Tracking Collar

A female wolf, nicknamed Carlina, was found, unable to move, in Lisciano (Rieti municipality) on 20 July 2013. Carlina was admitted to the rehabilitation centre of “Parco Faunistico Piano dell’Abatino” (Figure 2), located in the municipality of Poggio San Lorenzo, where she was treated and rehabilitated for 11 days. Given the conditions of initial immobility, Carlina was kept inside a cage 1 m wide, 2 m deep and 1 m high. According with the veterinarian judgment, given the state of asthenia, Carlina remained inside the cage until healing and releasing, to avoid the stress that could be caused by the recapture required if moved into a larger enclosure. Immediately after the check-in, Carlina was moved to a local specialized veterinary clinic for X-rays and neurological examinations. On the basis of tooth wear [21], the veterinarian estimated Carlina was between 4 and 6 years old. These exams revealed the absence of trauma and the occurrence of nerve endings’ acute inflammation, which was probably caused by a tick-borne infectious disease. Accordingly, Carlina was immediately subjected to antibiotic and corticosteroids therapy, which continued for 7 days. During the first two days, food and fluids were infused through an intravenous line. From the third day, Carlina started eating about 1 kg of fresh meat per day, and after 5 days of treatment she was able to sit up showing clear signs of healing. Direct interactions with persons were limited to three occasions/day (treatments and feeding) for approximately 1 h overall exposition time. According to the veterinary judgment, on 28 July 2013, the disease was resolved and 1 August, Carlina was released at 5 p.m. in the same area where she was found (Figure 2).

Before the translocation from the rehabilitation centre to the release area, Carlina was tagged with a Lotek GPS tracking collar (Lotek Wireless Inc., Newmarket, ON, Canada; www.lotek.com) (Figure 3). The veterinarian applied the device with the help of a single operator covering the wolf’s head with a blanket, avoiding muzzle and anaesthesia. The collar was set to drop off when the battery charge level was at 20% and programmed to acquire location every 30 min until 8 August, and every 3 h for the following days. Temporal (date and time) and spatial data were retrieved from the GPS collar through the Global System for Mobile Communications (GSM). The GPS receiver of the collar stopped working on 26 November 2013, and the signal was definitively lost.

For the study of daily movements, we have defined daytime as the period (expressed in hours) of visibility, including twilight before dawn and after sunset and nighttime as the period in which there is no light at all. According to Effemeridi Aeronautiche [22], we adopted fifteen hours a day for August, thirteen and a half hours a day for September, eleven hours and forty-five minutes a day for October and eleven hours a day for November.

### 2.3. Data Analysis

Overall, 2070 locations were acquired and projected in ArcMap 10.3 [23]. The original locations dataset was preliminarily filtered to avoid spatio-temporal overlapping between consecutive positions that could generate calculation errors on trajectory [24]. The filtering step generated a dataset of 870 valid locations. The home ranges and movement patterns (distance and speed) were calculated with ArcMet 10.3.1 [24]. Home ranges were calculated overall and separately for the summer season (1 August–30 September) when wild boar drive hunting is restricted and the autumn season (1 October–26 November) when this hunting was allowed, with the minimum convex polygon (MCP 95%) [25]. For the overall MCP, the land cover and use were determined by intersecting the polygon with the regional land-use map (Lazio Regional Administration 2010) [26,27].

Successive GPS locations were connected through the trajectory path tool. The output of this tool associates the distance (m) and the movement rate (MR = km/h) at each straight line between consecutive locations. The MR parameter approximates the wolf Carlina’s movement rate in its territory, but not as commonly as intended speed. This parameter was calculated on trajectory segments between thirty minutes and three hours, including resting phases and short and slow movements, probably due to being vigilant or engaging in feeding activities. Differently, the speed (km/h) parameter was calculated by excluding resting phases and limiting the analysis to the straight lines connecting fixes recorded during the first eight days of the summer (1–8 August), with a maximum of one fix every 30 min. Line segments connecting successive locations were classified considering the season and the phase of the day (nighttime, daytime). Daily distance travelled (DDT), movement rate (MR) and speed were then calculated separately for each season and phase of the day. In addition, the tortuosity of the seasonal movement pattern was assessed as the ratio of the sum of the straight lines in each seasonal trajectory to the distance between the first and the last point of the same trajectory [24].

Three types of non-active/active behaviours were also computed by season and phase of the day: (a) resting, in the case of displacements <50 m in one hour; (b) short displacement, between 50 and 500 m in one hour; (c) long displacement, larger than 500 m in one hour.

Statistical analyses were performed with R. The Kruskal–Wallis test was used for the variance analysis and computed with the R function kruskal.test {kruskal.test}. For pairwise comparisons, the Mann–Whitney U Test was computed by means of the R function wilcox.test {wilcox.test} [28]. A 5% significance level (*p* ≤ 0.05) was accepted for all the statistics.

## 3. Results

Carlina did not show dispersal behaviour—emigration, transience and settlement—in a new area, and overall moved within an MCP (95%) of 6528.96 ha (Figure 4), where almost 90% consisted of forests (74.92%), natural pastures and grassland (13.03%) (Table 2).

In the summer and the autumn, seasonal MCPs (95%) were 6458.68 ha and 6700.8 ha, respectively (Figure 5).

The maximum daily distance travelled by Carlina was similar in the two observation periods: 12,818.0 m/day during the summer and 12,623.9 during the autumn. Mean (±SE) daily distance travelled was 2062.38 ± 276.48 m in the autumn and 3150.04 ± 341.31 m in the summer with significant differences between seasons (U = 3008.0, *p* < 0.01).

Mean (±SE) diurnal and nocturnal DDT and MR were reported in Table 3. Carlina performed a nocturnal displacement of 4409.4 ± 617.5 m in the summer and 3684.8 ± 468.1 m in the autumn, with no differences between seasons. Diurnal displacements in the summer (2239.0 ± 329.0 m) were significantly larger than in the autumn (595.9 ± 110.3 m). However, differences between nocturnal and diurnal displacements were significant in both seasons (Table 3).

The mean rate of movement (MR) varies between 0.10 ± 0.01 km/h during the daytime in the autumn and 0.79 ± 0.1 km/h during the nighttime in the summer. Differences in MR between the nighttime and daytime are significant in both seasons, while differences between seasons are significant only in the daytime. The movement pattern was much more tortuous in the autumn (tortuosity = 177.98) than in the summer (tortuosity = 50.93).

During the summer, Carlina moved with speeds ranging between 0.1 and 2.73 km/h in the daytime and between 0.13 and 3.64 km/h at nighttime (Table 4). Differences between daytime and nighttime were not significant.

During the summer, when hunting is banned, Carlina spent 36.19, 33.15 and 30.66% of the daytime in resting, short displacements and long displacements, respectively (Figure 6). Therefore, the activity rate was high also during the daytime, with peaks close to dawn and dusk. In the nighttime, these percentages were 16.33%, 15.35% and 68.32%, respectively.

During the autumn, when wild boar drive hunting is in progress, Carlina spent 58.44, 29.53 and 12.02% of the daytime in resting, short displacements and long displacements, respectively. In the nighttime, these percentages were 17.64, 17.64 and 64.72% (Figure 6). In this period, Carlina frequented the hunting grounds at nighttime only and normally at least two days later than the last hunting actions. On only two occasions was Carlina located in the proximity of a drive hunting area the nighttime following a hunting action.

## 4. Discussion

The estimation of Carlina’s home range (65.29 km^2^) is smaller than that reported in the literature for central (137–294 km^2^) and southern Europe (120–200 km^2^) [29,30]. This could depend on the calculation with locations recorded only during summer and autumn, due to the subsequent loss of the GPS signal. Furthermore, high forest cover, availability of refuge areas and prey abundance influence wolves’ home ranges, reducing space need [20,31], and the study area is characterized by a widespread availability of these resources.

Consistent with the home range, the daily distance travelled by Carlina (0.6–4.4 km/day) is also similar to the values reported in the literature for a sub-adult female by Kolenosky and Johnston (0–5.6 km/day) [32], in the range reported by Kusak et al. [33] (0–13.2 km/day) for an adult and a sub-adult female, and by Ciucci et al. [30] for a rescued male of the Apennine area (0.3–11.5 km/day). It is lower than what Jedrzejewski et al. [34] recorded in Poland (6.6–51.3 km/day) with V.H.F. telemetry of three non-breeding females. Although these measurements should be considered underestimations of the real distances travelled [33], due to the time lapse between successive locations, short daily distances are usually related to a high abundance of preys [32]. Accordingly, the study area is characterized by abundant ungulates populations. Furthermore, Carlina territory intersects about 3116 ha of wild boar hunting areas, where these ungulates are frequently injured and die after a short time [35,36]. The presence of a protected area (hunting banned area)—the Terminillo wildlife oasis (Figure 1 and Figure 5)—could generate, especially during the hunting season (autumn–winter), concentration in ungulates due to the refuge effect [37,38,39], favouring predator–prey encounter probability [20]. Accordingly, Carlina was frequently located along the border between Terminillo oasis and the neighbouring hunting zones (Figure 5).

Carlina did more protracted displacements during the nighttime than during the daytime in both autumn and summer. According to Kusak et al. [33], more protracted displacements are associated with higher movement rates and speeds.

The percentages of time spent in the different non-active or active behaviours (Figure 6) revealed that during the summer, Carlina was also active during the daytime, at dusk and dawn in particular, as also reported by Theuerkauf et al. [40]. Differently, during the autumn, activity concentrates on the nighttime when hunting was in progress. At the same time, during the daytime, Carlina showed only resting phase and short displacements, probably associated with vigilance.

As previously recorded on wolves not subjected to captivity and forced contact with humans, Carlina’s movement patterns and circadian rhythms highlighted evident complementarity with human activities, concentrating the activity phases during the nighttime when human presence is low [20,31,41]. Accordingly, during the autumn, when the hunting season is in progress, Carlina only uses hunting areas at nighttime. Considering a standard utilization of the hunting areas as in the huntable days scheduled by the Lazio region (Wednesday, Saturday and Sunday of each week since 2 October 2013), according to Viola et al. [20], Carlina would use the wild boar drive hunting areas at least two days after the last hunting actions, except for two occasions in which the wolf was located in the proximity of a hunting area the nighttime of the same hunting day.

Previous work, which involved wild wolves that were not forced into close contact with humans [20,31,42,43,44], reported that these animals avoided anthropic disturbance and limited the attendance of areas or infrastructures frequently used by humans (cultivated lands, suburbs or areas close to roads) in a complementary way. Consistent with these results, Carlina selected and used habitats characterized by a low degree of anthropization (2.15%, urban areas; 1.34%, roads; 4.21%, cultivated areas) and high wilderness, with almost 90% consisting of natural pastures (13%), primarily used for extensive livestock farming and forests and shrubs (79%) (Table 2).

Furthermore, the attendance of anthropized contexts (i.e., cultivated lands and pastures near human settlements, as well as hunting areas in autumn) was limited at nighttime or at the non-use period by persons anyway, confirming a marked complementarity.

Since human-habituated wolves, often as the result of food-conditioning, shift their home range close to human settlements and anthropized area in an opportunistic way, adopting circadian rhythms that are noncomplementary but similar or concomitant to the humans’ ones [45,46], Carlina did not show signals of human habituation after 11 days of veterinary isolation, treatment and non-agonistic experience with humans.

## 5. Conclusions

In this case study, the wolf has attended wild context (79% wooded), avoiding anthropized areas and maintaining evident complementarity, with respect to human activities, as confirmed by other authors [20,31,41]. Indeed, she was constantly more active during the nighttime in both seasons and drastically limited long diurnal displacements (12.2%) during the hunting season (autumn), while during the summer, protracted displacement amounted to 30.66% of the total diurnal active time. No signs of habituation to humans were detected, but this is only a first case study involving one wolf for a very short period of veterinary isolation and treatment. Therefore, this study does not aim to be conclusive but a stimulus for further research and a call for widespread data sharing at national and international scales. The interest should not only be aimed at increasing the overall sample size but at obtaining more test groups—i.e., time spent ex situ, sex and age class—in order to identify any threshold values beyond which it might be appropriate to avoid reintroducing the animal into the wild.

## Figures and Tables

**Figure 1 animals-12-03495-f001:**
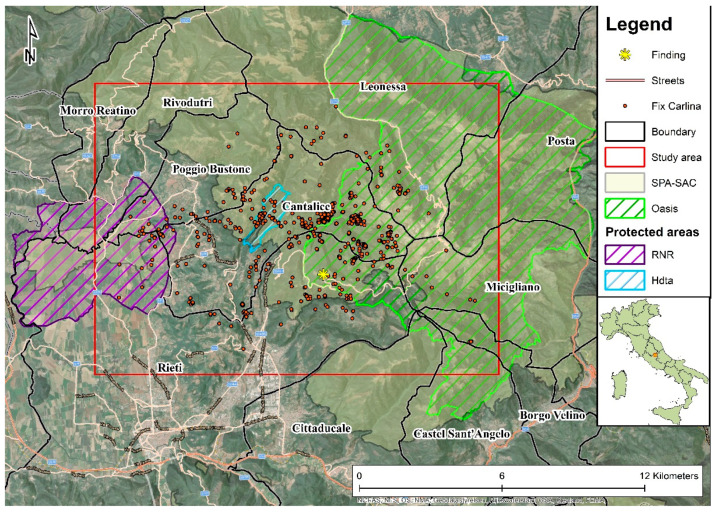
Study area, location of finding/releasing and Carlina locations (Fix). Boundary = municipalities boundaries; SPA = special protection area (EU Birds Directive 2009/147/EC); SAC = special area of conservation (EU Habitats Directive 92/43/EEC); RNR = regional natural reserve “Laghi Lungo e Ripasottile”; Oasis = Terminillo wildlife oasis; Htda = hunting dog training area. The locality where Carlina was found, unable to move, on 21 July 2013 is Lisciano (RI).

**Figure 2 animals-12-03495-f002:**
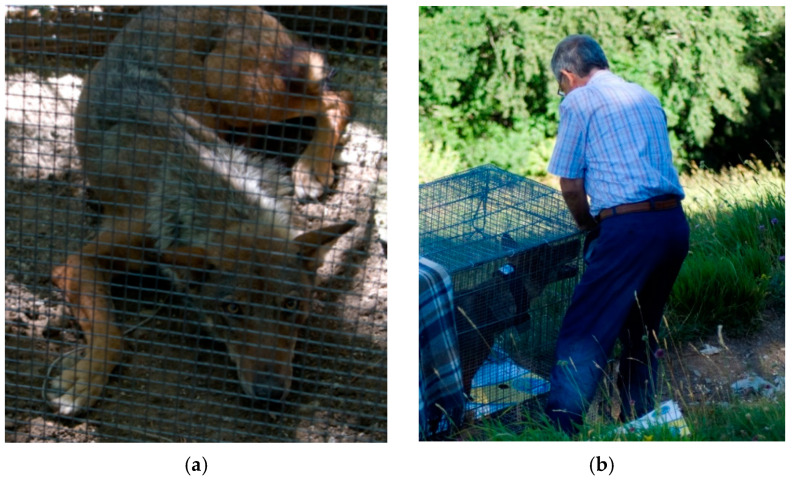
Carlina the wolf: (**a**) in the rehabilitation centre of “Parco Faunistico Piano dell’Abatino”; (**b**) immediately before the release in the area where she was found.

**Figure 3 animals-12-03495-f003:**
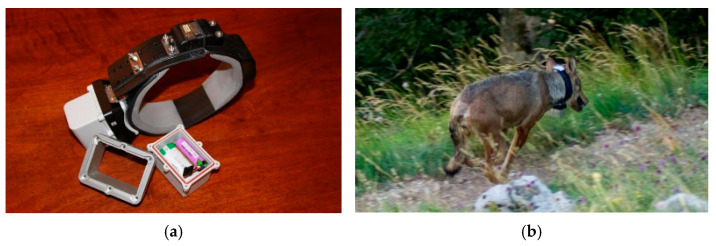
GPS tracking collar (Lotek Wireless Inc.): (**a**) GPS collar before being applied to the wolf, (**b**) Carlina wolf equipped with the GPS collar and photographed immediately after the release.

**Figure 4 animals-12-03495-f004:**
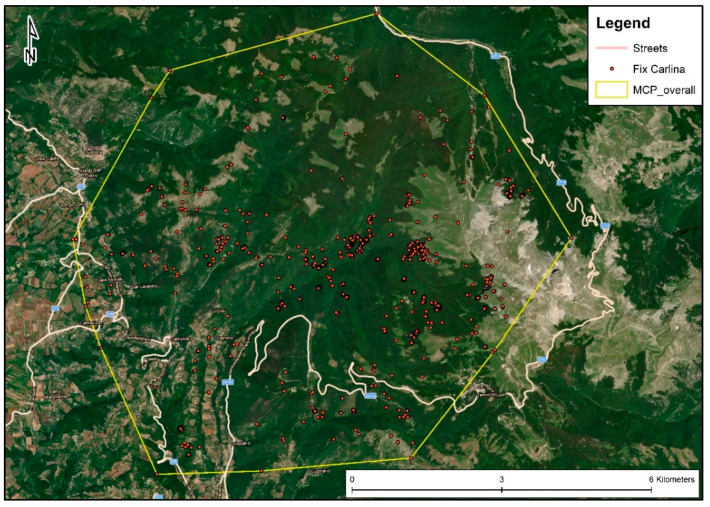
Representation of the minimum convex polygon (MCP) of the female wolf nicknamed Carlina generated starting from the GPS locations (Fix Carlina) recorded in the period between 1 August and 26 November 2013 (MCP overall).

**Figure 5 animals-12-03495-f005:**
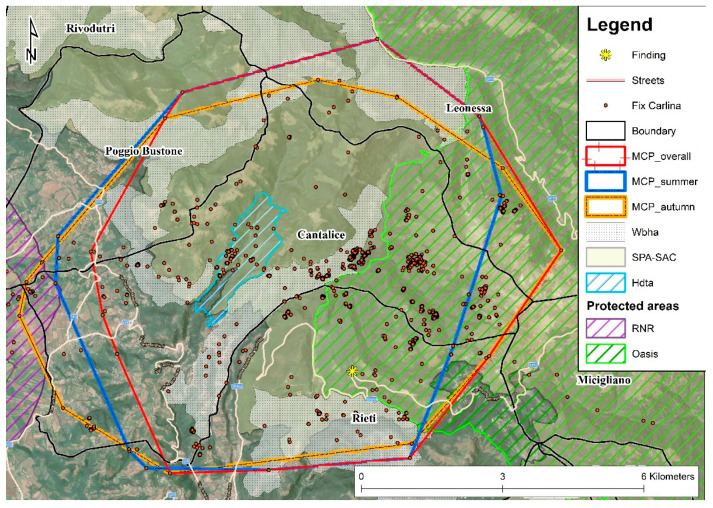
Study area, location of finding and releasing, overall and seasonal minimum convex polygons (95%) and Carlina wolf locations. MCP_summer = during the summer the wild boar drive hunting is banned; MCP_autumn = during the autumn this hunting is in progress. Wbha = wild boar drive hunting areas; Htda = hunting dog training areas; Oasis = Terminillo wildlife oasis; RNR = regional natural reserve “Laghi Lungo e Ripasottile”.

**Figure 6 animals-12-03495-f006:**
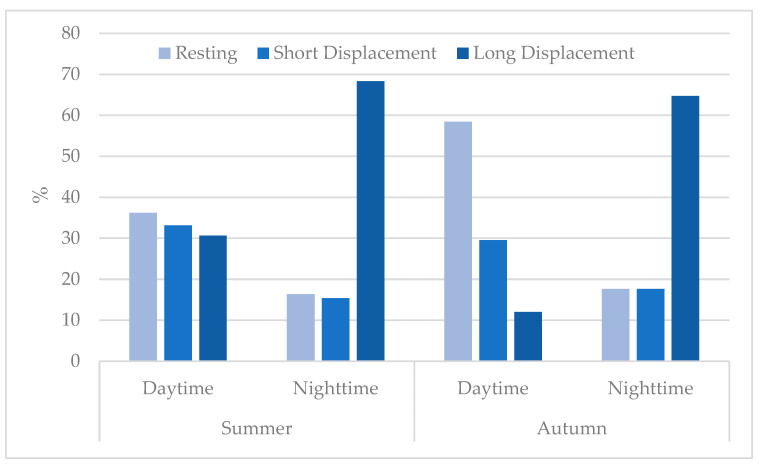
Percent (%) of time spent in three types of non-active/active behaviours computed by season and phase of the day: (a) resting, in the case of displacements <50 m in one hour; (b) short displacement, between 50 and 500 m in one hour; (c) long displacement, larger than 500 m in one hour.

**Table 1 animals-12-03495-t001:** Land-covers of the study area.

Category	Typology	Area (ha)	%
Broad-leaved forests	Beech, hop, hornbeam, oak and other mixed woods	11,510.48	55.83
Coniferous forest	Black pine	671.72	3.26
Cultivated lands	Permanent crops and arable lands	3137.92	15.22
Fruit trees	Chestnuts	243.41	1.18
Open areas	Natural grassland and pastures	2755.63	13.37
Scrubland	Bushes and shrubs	119.60	0.58
Principal road	Paved roads	854.42	4.14
Secondary roads	Gravel roads	269.59	1.31
Urban areas	Settlements and human activities	923.11	4.48
Water bodies	Lake and rivers	129.27	0.63
Total		20,615.14	100.00

**Table 2 animals-12-03495-t002:** Land covers of the overall minimum convex polygon (MCP).

Category	Typology	Area (ha)	%
Broad-leaved forests	Beech, hop, hornbeam, oak and mixed woods	4733.30	72.36
Coniferous forest	Black pine	167.57	2.56
Cultivated lands	Permanent crops and arable lands	275.12	4.21
Open areas	Natural grassland and pastures	852.54	13.03
Scrubland	Bushes and shrubs	280.45	4.29
Principal road	Paved roads	31.68	0.48
Secondary roads	Gravel roads	56.80	0.87
Urban areas	Settlements and human activities	140.40	2.15
Water bodies	Lake and rivers	3.27	0.05
Total		6528.96	100.00

**Table 3 animals-12-03495-t003:** Nocturnal and diurnal daily distance travelled and movement rate during the summer and the autumn. Kruskal–Wallis ANOVA tested differences for multiple comparisons and Mann–Whitney U Test for pairwise comparisons.

	Daily Distance Travelled (m)				Movement Rate (km/h)			
	Daytime	Nighttime	U Test		Daytime	Nighttime	U Test	
Season	Mean ± SE	Mean ± SE	U	*p*	Mean ± SE	Mean ± SE	U	*p*
Summer	2239.0 ± 329.0	4409.4 ± 617.5	482.0	<0.01	0.25 ± 0.04	0.79 ± 0.1	10,823.0	<0.001
Autumn	595.9 ± 110.3	3684.8 ± 468.1	315.0	<0.001	0.10 ± 0.01	0.53 ± 0.1	3074.0	<0.001
U	651.0	672.0	/	/	17,036.0	6134.0	/	/
*p*	<0.001	0.311	/	/	<0.001	0.079	/	/

**Table 4 animals-12-03495-t004:** Mean (mean ± SE) nocturnal (nighttime) and diurnal (daytime) speeds in summer. Kruskal–Wallis ANOVA tested differences.

	Speed (km/h)					
	Daytime		Nighttime		Kruskal–Wallis	
Season	Mean ± SE	Range	Mean ± SE	Range	H	*p*
Summer	1.30 ± 0.16 (28)	0.10–2.73	1.37 ± 0.11 (66)	0.13–3.64	0.11	0.74

## Data Availability

The data presented in this study are available on request from the corresponding author.

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
