# Peer review of "Does the Wolf (Canis lupus) Exhibit Human Habituation Behaviours after Rehabilitation and Release into the Wild? A Case Report from Central Italy"

_animals, 2022, doi:10.3390/ani12243495_

Round 1
Reviewer 1 Report
Dear authors, your work is important especially in light of Italian wolf population' expansion in Italy that is also happening in densily-populated areas such as the Po plain. You case study, despite the very short period of captivity, is encouraging and more research should be done in this direction.
I attach here the comments and suggestions that come from my revision.
best,
the reviewer

Author Response
Dear Reviewer,
we appreciate the time and effort that you dedicated to providing feedback on our manuscript and are grateful for the insightful comments on and valuable improvements to our paper.
We have incorporated most of the suggestions you made. Those changes are highlighted within the manuscript. Please see the attached pdf file for a point-by-point response to your comments and concerns and the revised manuscript file with tracked changes.

Reviewer 2 Report
The overall topic of this project is an interesting and suitable one to address. I think the justification, method and results of the study are clearly explained. Though I do feel that the term captivity is not quite appropriate here and instead you are referring to veterinary isolation and treatment rather than a captive setting where animals may have the space and social grouping to perform a more normal repertoire.
Can you please clarify if the cage described was the only enclosed environment for the 11 days duration or was this just the initial placement of the animal during recovery?
How was the collar attached, was the animal anesthetised, if so what was the procedure for this and was this extended beyond that needed for regular treatment?
Was the collar set to drop off after a set period of time?
The aspect of attaching collars should have been ethically reviewed, especially as this could have impeded the chance of reforming bonds with the pack and survivability.
The wolf is not an inanimate object, and so should be referred to as she not it.
Why are some p values given as < and some stated as 0, it is very rare to have p values that are exactly 0.
Was any observations of Carlina made visually after release. Did she socialise with other wolves?
Could the shorter travel distances be explained by lack of physical condition experienced after treatment?
Author Response

(The authors gave the same response as above.)

Reviewer 3 Report
Thank you very much for an opportunity to review the paper. Human influence on wildlife is an important issues and has gained more and more attention nowadays. Every paper on this topic is very precious and gives a glance at how anthropopressure affects animals' behaviour. This work shows wolf behaviour after rehabilitation and releases into the wild (a case from central Italy). The research is very interesting, and I have minor comments, all listed in the manuscript (attached). I have two major comments: 1. Why the Authors used shortcuts everywhere? I would recommend using the whole words. 2. The whole discussion is hard to follow, mostly because of the too-long sentences, but also because I can’t recognize where the Authors describe their results and where they wrote about literature. Moreover, some of the results described in discussion I cannot find in results.
I wish the Authors all the best. Please see attached file.

Author Response

(The authors gave the same response as above.)
